# Focal Mechanisms and Stress Field Characteristics of Microearthquakes in Baihetan Reservoir in the Downstream Area of Jinsha River

**Wei Guo [1,2] and Cuiping Zhao [1,*]**

1   Institute of Earthquake Forecasting, China Earthquake Administration, Beijing 100036, China
2   Seismological Bureau of Inner Mongolia Autonomous Region, Hohhot 010010, China
*   Correspondence: zhaocp@ief.ac.cn

**Abstract:** The Baihetan Reservoir was impounded on 6 April 2021, after which the water level rose significantly. Notably, after one week of impoundment, microseismic activities were prominent around the reservoir area, which was highly associated with the water level change. From 6 April 2021 to 31 December 2021, over 7000 microearthquakes were recorded by the seismic stations in the vicinity of the reservoir, including 12 $M_L > 3$ events. The maximum was the 21 December 2021 $M_L 3.9$ earthquake in Qiluogou town, Sichuan. The post-impoundment seismic events were clustered in Hulukou town in the Qiaojia Basin, with an overall "Y-shaped" pattern. In this study, taking advantage of the high-frequency waveform matching approach, the pre- and post-impoundment focal mechanism solutions totaling 207 $M_L > 2$ earthquakes are successfully obtained. The impoundment-induced stress change is analyzed, and the iterative joint inversion method is used to invert the stress field. Major results and conclusions include the following: (1) After impoundment, the number of normal fault earthquakes remarkably increased in the reservoir area; (2) Impoundment has led to a vertical compressive stress field and horizontal tensile stress field in the area where microearthquakes occurred. It is necessary to pay close attention to possible moderate-to-strong earthquakes in the future.

**Keywords:** Baihetan Reservoir; reservoir-induced earthquakes; focal mechanism; stress field

## 1. Introduction

The downstream area of the Jinsha River is located on the eastern margin of the Sichuan-Yunnan block, and is characterized by complex geological structures and frequent seismic activity. The regional topography is high in the west and low in the east, producing a huge drop and, hence, an abundant hydraulic energy resource. From south to north, 4 cascade hydropower stations (Wudongde, Baihetan, Xiluodu, Xiangjiaba) were successively constructed (Figure 1). In particular, the Baihetan station is situated at the junction zone of several large geological structures with different strikes. It possesses the largest reservoir storage capacity and the highest cascade benefit among the four stations, and is the second-largest hydropower station in the world (ranked only behind the Three Gorges station). This station has a height of 289 m, a storage capacity of up to $206.27 \times 10^8$ m$^3$, and an installed gross capacity of up to 1600 million kilowatts (kw). The Xiangjiaba, Xiluodu, Wudongde, and Baihetan Reservoirs began their initial impoundment in October 2012, May 2013, January 2020, and April 2021. Notably, a process of seismicity enhancement dominated by microearthquakes appeared after the impoundment of the Xiluodu Reservoir [1,2]. In this study, we obtained high-quality observational seismic data 5 years before and 9 months after the impoundment of the Baihetan reservoir through the seismic arrays continuously deployed in this region since 2016. A concentration of small earthquakes occurred in Hulukou town immediately after the impoundment. Baihetan Reservoir was impounded on 6 April 2021, after which the water level rose significantly.

Notably, after impoundment, microseismic activities were prominent around the reservoir area, highly associated with water level change. From 6 April 2021 to 31 December 2021, a total of over 7000 earthquakes were recorded by the seismic stations in the vicinity of the reservoir, including 12 events with a magnitude greater than 3. The earthquakes were dominantly clustered in Hulukou town and distributed along the main Jinsha River and the Heishuihe tributary [3].

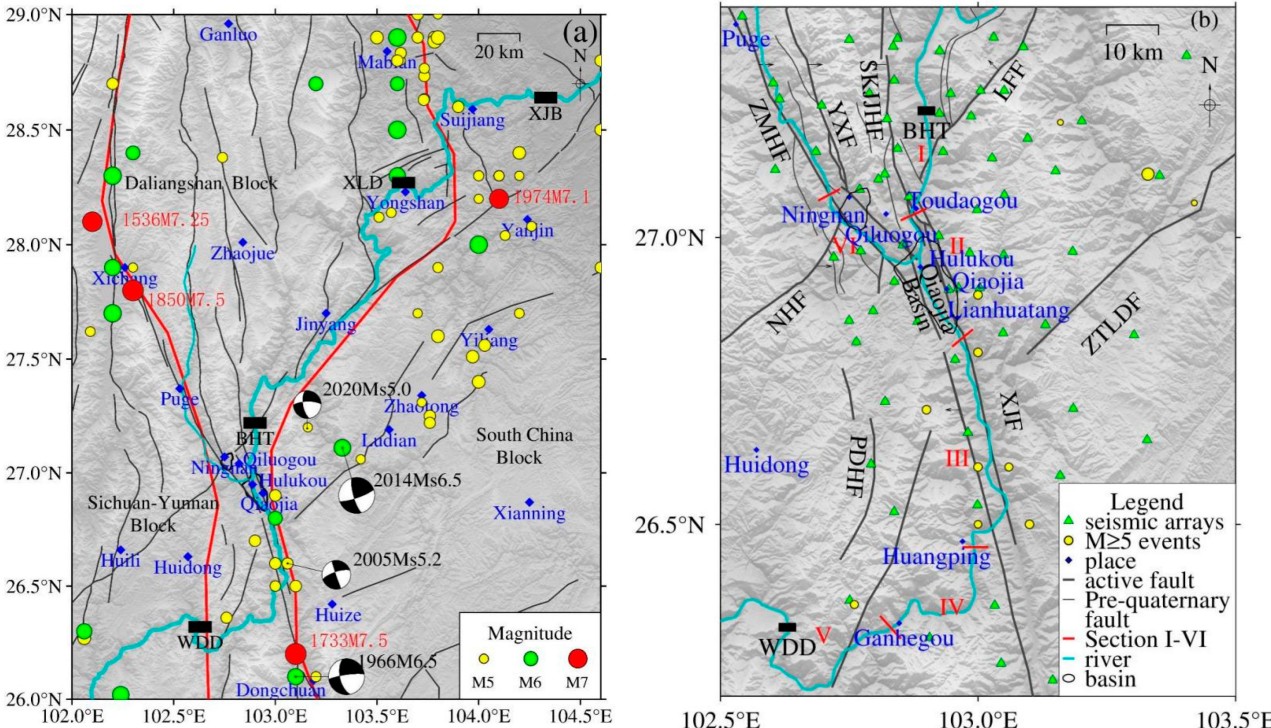

**Figure 1.** Tectonic background, historical earthquakes and distribution of seismic stations. (**a**) Tectonic background and historical earthquakes in the downstream area of Jinsha River; (**b**) Tectonic background, historical earthquakes, and distribution of seismic stations in the Baihetan Reservoir. Black rectangles represent hydropower stations. XJB: Xiangjiaba hydropower stations; BHT: Baihetan hydropower stations; XLD: Xiluodu hydropower stations; WDD: Wudongde hydropower stations. Black lines represent faults, obtained from activefault-datacenter.cn. LFF: Lianfeng fault; SKJJHF: Sikaijiaojihe fault; YXF: Yuexi fault; ZMHF: Zemuhe fault; NHF: Ninghui fault; PDHF: Puduhe fault; XJF: Xiaojiang fault.

Investigating the seismic focal mechanism and tectonic stress field before and after reservoir impoundment is essential for a more thorough understanding of the various characteristics of seismic type and stress state, providing significant evidence for further probing the mechanism of reservoir-induced earthquakes and analyzing the risk of larger earthquakes in the future. Over the past decades, numerous studies regarding the focal mechanism of earthquakes in typical large reservoirs have been conducted. For example, through studying the 19 March 1962 M6.1 earthquake in the Xinfengjiang Reservoir, Wang et al., (2002) [4] proposed that strike-slip was the main mechanism for the seismogenic faults of the mainshock and aftershocks within 18 months. The mechanism altered to dip-slip after 58 years, as indicated by He et al., (2018) [5] after calculating the focal mechanism of *M*s > 1.5 earthquakes in the same region. They also suggested that the main type is still a strike-slip mechanism outside the reservoir area. The long-lasting diffusion effect increased the pore pressure, resulting in deeper earthquakes and hence the change of seismic type. Yao et al., (2017) [6] analyzed the variation characteristics of seismicity and related focal mechanisms throughout different impoundment periods of the Three Gorges Reservoir. They suggested that seismicity was closely associated with water level change.

During the impoundment period with 135 m, 156 m, and 175 m water levels, strike-slip and normal mechanisms were the most dominant. In addition, the P- and T-axes were discretely distributed, inconsistent with the regional tectonic stress field. By contrast, multiple larger earthquakes occurred in the later stage of the 175 m impoundment, and the P- and T-axis distribution was accordant with the stress field.

It was suggested that the small regional earthquakes were triggered by the reservoir, while the large earthquakes were controlled by regional faults and tectonic stress fields. Similar investigations in the Longtan Reservoir in Guangxi [7,8] suggested a variety of earthquake types in the early stage of impoundment. After 51 months, earthquakes were primarily seen in the shallow layers and were mostly related to reverse faults. The difference of seismicity and seismic properties between shallow and deep layers can be attributed to the difference in their intrinsic features, such as the tectonic stress field, rock mechanics properties, and permeability. The study area of this paper, which is the downstream area of the Jinsha River, has also been investigated by some researchers. Xiluodu Reservoir was impounded on 4 May 2013. As of 31 October 2013, Diao et al., (2014) [2] obtained the focal mechanisms of over 700 earthquakes before and after the impoundment of the Xiluodu Reservoir (2007–2013). They reported that the focal mechanisms of small earthquakes at the initial stage of impounding were characterized by complicated spatial fault plane orientation, various fracturing types, and unstable stress states. Duan and Zhao (2019) [9] further calculated the focal mechanism solutions of earthquakes (2016–2018) in this area, obtained the maximum principal compressive stress $\sigma_1$ axis and the maximum tensile stress $\sigma_3$ axis, and compared them with the results calculated by Diao et al., (2014) before and at the initial stage of impounding. The results showed that the spatial distribution of the stress axis is closer to that before impoundment. Therefore, Duan and Zhao (2019) concluded that as of 2018, the stress state had gradually recovered to the pre-impoundment state. The pre-impoundment focal mechanism in Baihetan and Wudongde Reservoirs is dominated by the strike-slip type.

The seismicity and focal mechanism solution before and after the water impoundment of Baihetan Reservoir provided an important example for investigating reservoir-induced earthquakes in Alpine canyon reservoir regions. Based on the precise earthquake location and seismicity analysis, together with the assistance of waveform data collected by the continuously completed seismic array since 2013, the focal mechanism of small earthquakes in the Baihetan Reservoir area was successfully obtained using the waveform matching method. Furthermore, the pre- and post-impoundment seismicity features and the corresponding variation characteristics of the epicenter and stress field were analyzed to provide data for determining the future large earthquake risk and other relevant studies [10].

## 2. Tectonic Background and Data

The Baihetan Reservoir in the lower reaches of the Jinsha River is located in the transitional zone extending from the strongly uplifted Qinghai Tibet Plateau and the eastern edge of the Western Sichuan Plateau to the Yunnan Guizhou Plateau and Sichuan Basin. The dam site is located east of the Kangdian axis within the Yangtze Huaihe Platform [11]. The left bank of the dam site is Ningnan County, Sichuan Province, and the right bank is Qiaojia County, Yunnan Province. The regional geological structure is complex. Deep and large fractures have developed, and some intersect the Jinsha River. The regional large fault east of the Zemuhe-Xiaojiang fault zone, driven by the northwest-southeast principal compressive stress field, is characterized by the northwest to southeast thrusting nappe, while the strike-slip movement occurs under the traction of the left lateral strike-slip movement of the Xiaojiang fault zone. The blocks enclosed by the regional fault zone, driven by the Sichuan Yunnan rhombic block wedged in the south-southeast direction, show different degrees of rotation, resulting in the Ningnan Basin and Qiaojia Basin. The Zemuhe fault strikes north-northwest, extending from the north of Xichang to the Ningnan and Qiaojia, and the fault dips northeast with a high dip angle. The Lianfeng-Qiaojia fault developed along the axis of the Lianfeng anticline and extends northeast along

Qiaojia-Dazhai Lianfeng-Yanjin, and the fault dips northwest with a high dip angle. The Sikaijiaojihe fault strikes nearly north-south, and it is in a right-step oblique sequence along Sikai-Toudaogou-Jinshajiang and Xiaojiang fault zones. The Yuexi fault intersects with the Zemuhe fault in the northwest area of Ningnan town, both of which are sinistral strike faults.

The Zemuhe fault zone is characterized by obvious segmentation. Around Ningnan, it is rifted into a basin. Each segment is composed of several small feather-like structures, and the southeast part, the Ningnan-datong fault, shows frequent seismic activity. The overall strike and dip of the Ningnan-datong fault are N40°W and NE, with a dip angle of more than 60° and a length of about 20 km. The reservoir water is closely connected with the Ningnan-datong fault. The Xiaojiang fault extends from Qiaojia to the south along the Jinsha River and Xiaojiang River valley, trending north-northwest and dipping to the west in the section of Qiaojia, and turning to nearly north-south in the southeast of Dongchuan. The Qiaojia Basin along the Xiaojiang fault extends along the Jinsha River valley, 13 km long from north to south and 3–5 km wide. Hulukou Town, located at the junction of the Zemuhe fault, Sikaijiaojihe fault, and the north section of the Xiaojiang fault zone, is also where the Jinsha River and the northwest branch of the Heishui River converge. Due to the relatively open terrain in the valley, it is completely submerged after impounding. The length of the Zemuhe fault zone inundated from Hulukou along the Heishuihe branch is about 20 km, and the length of the Xiaojiang fault zone inundated by reservoir water from the Qiaojia Basin to the tail of the Xiaojiang branch is about 50 km. The Xiaojiang fault zone and Jinsha River experience historically frequent seismic activity. Since 1770 BC, there have been 9 earthquakes above $M$s5.0 in the reservoir area. The Menggu $M$s6.0 earthquake of 15 May 1935 in the south of Qiaojia was only 10 km from the reservoir area. The 3 August 2014 Ludian $M$s6.5 earthquake and 18 May 2020 Qiaojia $M$s5.0 earthquake that occurred in recent years were also near the reservoir area (Figure 1).

In this study, using the observational data (2013.8.1–2021.12.31) recorded by the dense seismic array in the downstream area of the Jinsha River, the pre- and post-impoundment seismicity and focal mechanism solution features of earthquakes are analyzed in the vicinity of the Baihetan Reservoir area (26.2°~27.4° N, 102.5°~103.5° E). The seismic array consists of a total of 169 seismic stations, including 74 stations deployed by the research group of the present paper in the downstream area of the Jinsha River, 62 stations that belong to the Qiaojia seismic array deployed by the Institute of Geophysics, and 33 stations from the Sichuan, Yunnan, and Guizhou Regional Seismic Network. The seismometers are CMG-3ESPC, CMG-3T, CMG-40T, Trillium 120P, BBVS-60, GL-PCS60, etc. These instruments have a frequency band range of 50 Hz~60 s or 50 Hz~120 s, with a sampling rate of 100 Hz. The 63 stations from the Jinsha River dense array are distributed throughout the Baihetan Reservoir area (Figure 1), with a station interval smaller than 10 km and the ability to monitor $M_L$ < 1.0 microearthquakes. Such extensive and widespread station distribution has laid a solid foundation for studying the seismicity and related focal mechanisms before and after the Baihetan Reservoir impoundment.

After comprehensively analyzing the regional geological structure, lithology, hydrogeology, and seismicity, the main river channel (i.e., from the Baihetan Dam to the Wudongde Dam) was divided into five sections according to different risk levels of reservoir-induced earthquakes. Based on the observational results in this study, we added the sixth section (the Heishuihe tributary section), in which the increase in water level in the Jinsha River triggered several microearthquakes. Finally, six reservoir sections were selected for monitoring and investigation (Figure 1b).

## 3. Method and Parameter Setting

The seismic activity in the Baihetan Reservoir is dominated mostly by microearthquakes, and the magnitudes are generally less than $M_L$3.0. In this study, the high-frequency waveform matching method [12,13] is employed to obtain the focal mechanism solutions of these microearthquakes. This method realizes the maximum matching degree between the

seismic phases and amplitude of both observational and theoretical waveforms. Furthermore, the method constrains the matching degree using the P-wave first motion and the S/P amplitude ratio, thereby constructing the objective function that contains four types of constraints and searching for the optimal solution using the grid-searching approach. The Green function utilizes the discrete wavenumber method for calculation [14,15], and the objective function is as follows:

$$
\begin{aligned}
&\text{maximize}[\text{J}(x,y,z,dip,rake,ts)]\\
&= \sum_{n=1}^{N}\sum_{j=1}^{3}\left\{\alpha_1\max(\widetilde{d}_j^n \otimes \widetilde{v}_j^n) - \alpha_2||\widetilde{d}_j^n - \widetilde{v}_j^n|| + \alpha_3 f[pol(\widetilde{d}_j^n), pol(\widetilde{v}_j^n)] + \alpha_4 h[rat(\tfrac{S(d_j^n)}{P(d_j^n)}), rat(\tfrac{S(v_j^n)}{P(v_j^n)})]\right\}
\end{aligned}
\tag{1}
$$

where $\widetilde{d}_j^n$ represents the normalized data, and $\widetilde{v}_j^n$ is the normalized theoretical waveform. $\alpha_1 \sim \alpha_4$ are the weighting factors of each term. The optimal value is selected considering that each term cannot dominate the objective function. Specifically, the first term calculates the maximum cross-correlation coefficient between the normalized data and normalized theoretical waveform. The negative sign in the second term is used to minimize the amplitude difference. These two terms are not independent, and their combination can better constrain the waveform similarity. The third term calculates and determines if the P-wave first motion polarity of the observational data is consistent with that of the theoretical waveform. The fourth term measures the consistency of the S/P amplitude ratio between the observational waveform and the theoretical waveform. In addition, the weighting factors of the waveform, the P-wave first motion, and the amplitude ratio are selected as 4, 2, and 0.5, respectively. The matching frequency band is set as 2–4 Hz. During the inversion of focal mechanism solutions, a layered velocity model constructed by the artificial depth-measuring profile [16] is employed to compute the Green function.

## 4. Results and Analyses

### 4.1. Seismicity in the Reservoir Area

The Baihetan Reservoir started its impoundment on 6 April 2021, and the water level gradually increased from 658 m (Figure 2a). As of 13 April, the water level had reached 690 m (Figure 2b), and the number of microearthquakes dramatically increased from below 100 (Since 2016) to over 400 times monthly (Figure 2c). As of 30 September 2021, the water level had reached the maximum (816 m), and both the monthly and daily earthquake frequencies reached their peaks (Figure 2d). After that, with the gradual drop in the water level, earthquake frequency also showed strong fluctuation. As of 31 December 2021, 7401 above $M_L0$ earthquakes had been recorded by the authors of the present study, including 5400 $M_L0.0$–0.9 events, 1858 $M_L1.0$–1.9 events, 141 $M_L2.0$–2.9 events, and 12 $M_L3.0$–3.9 events. The largest one, the 21 December 2021 $M_L3.9$ earthquake, occurred in Ningnan, Sichuan.

The post-impoundment seismic events in the reservoir area were clustered in Hulukou town, 37 km away from the dam, with an overall "Y-shaped" pattern. The three seismic branches of the "Y" pattern lie along the Sikaijiaojihe and Jinsha River, the Zemuhe fault and Heishuihe tributary, and the northern section of Xiaojiang fault and the Jinsha River, with a length of about 30 km, respectively. After impoundment, the seismic events first occurred in the area of Hulukou town and then developed northward to section I (Figure 3b). The seismicity stopped about 7 km from the dam along the Sikaijiaojihe fault and the Jinsha River towards the dam, forming three small belts spreading north-northwest from the riverside. The southern section of the Sikaijiaojihe fault is divided into two branches, the western and eastern parts, both of which dip steeply to the west. The branch fault is distributed in the same direction. The focal depth of these belts is relatively shallow, concentrated at 0–8 km (Figure 3d). After one week of impoundment, earthquakes along the north section of the Xiaojiang fault and in the Qiaojia Basin occurred in section II (Figure 3b). The northern section of the Xiaojiang fault is hydraulically connected with the reservoir water. Several concealed water-filled fault zones parallel the Xiaojiang fault developed in the Qiaojia Basin. The focal depth of earthquakes in section II is deeper than

that of reservoir section I, concentrated at 0–10 km. It is characterized by the continuous deepening of the focal depth southward (Figure 3d). The seismic activity along the north section of the Xiaojiang fault and the Jinsha River towards the south stopped near the Ms6.0 earthquake in Qiaojia South Menggu on 15 May 1930. After one month of impoundment, earthquakes began to occur in section VI (Figure 3b). The earthquakes were distributed in belts along the Zemuhe fault or Heishuihe tributary (Figure 3d). The focal depth was concentrated at 0–9 km. The farther away from the Heishuihe tributary, the shallower the focal depth (about 3 km). The focal depth of earthquakes near the river and the fault was relatively deeper (about 8 km). The seismic activity along the NW Zemuhe fault and the branch direction of the Heishui River stopped near the south of Ningnan.

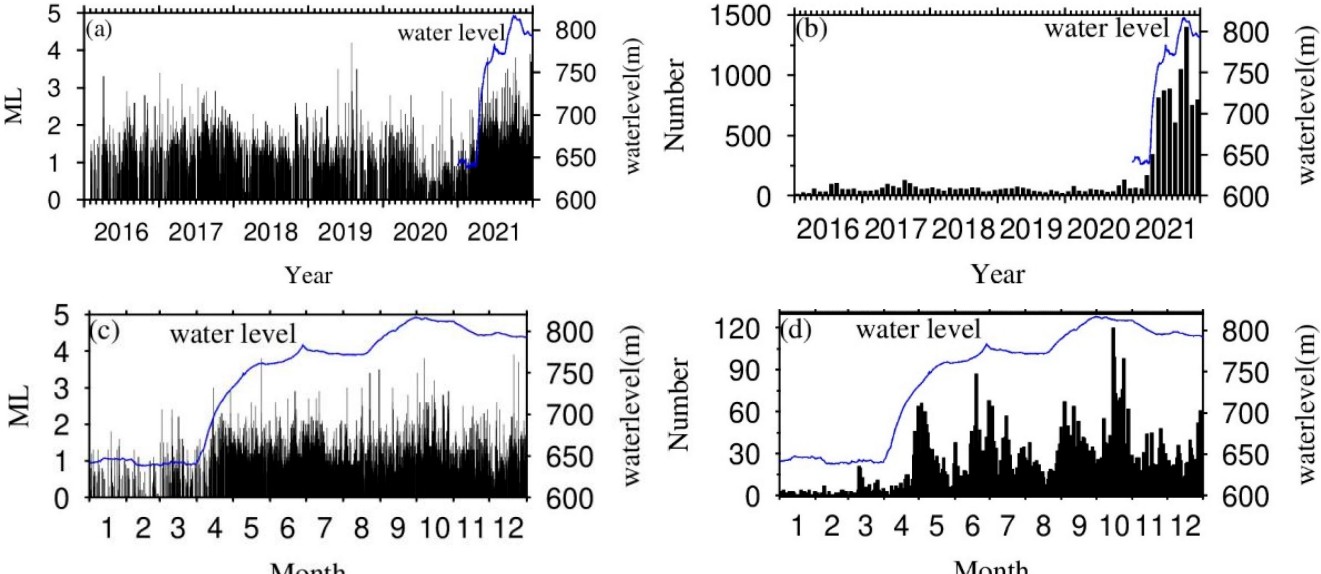

**Figure 2.** Magnitude, monthly frequency of seismicity, and water level sequence diagram in the Baihetan Reservoir. (**a**) Magnitude-time plot; (**b**) Monthly frequency-time plot; (**c**) Magnitude and water level (blue line) since January 2021; (**d**) Daily frequency and water level (blue line) of seismicity since January 2021.

*4.2. Focal Mechanism Solution*

The 207 $M_L \geq 2$ focal mechanism solutions before and after the impoundment of Baihetan Reservoir were obtained using the high-frequency waveform matching method. Figure 4 shows the waveform matching results of the 22 June 2021 $M_L$ 2.2 earthquake in Ningnan, Sichuan, and the 24 December 2021 $M_L$ 3.7 earthquake in Qiaojia, Yunnan. In this study, the classification method proposed by Zoback (1992) [17] was utilized (Table 1) to compile the focal mechanism solutions in the Baihetan Reservoir, according to the plunge of the P, B, and T-axes.

To evaluate the stability of the present study's results, the mean value and standard deviation of the first 100 best solutions of the two events were calculated (Figure 4) (Table 2). The results show the errors of strike, dip, and rake were all around 10°, consistent with the stability testing results of Li et al., (2011) [12].

Figure 5 displays the pre- and post-impoundment earthquake locations and the corresponding mechanism solutions of reservoir sections I, II, and VI. In general, normal fault-induced earthquakes are the most dominant after reservoir impoundment, which is similar to the feature that occurred in the downstream Xiluodu Reservoir during its water level increase stages. In the following, the focal mechanism features are analyzed section by section.

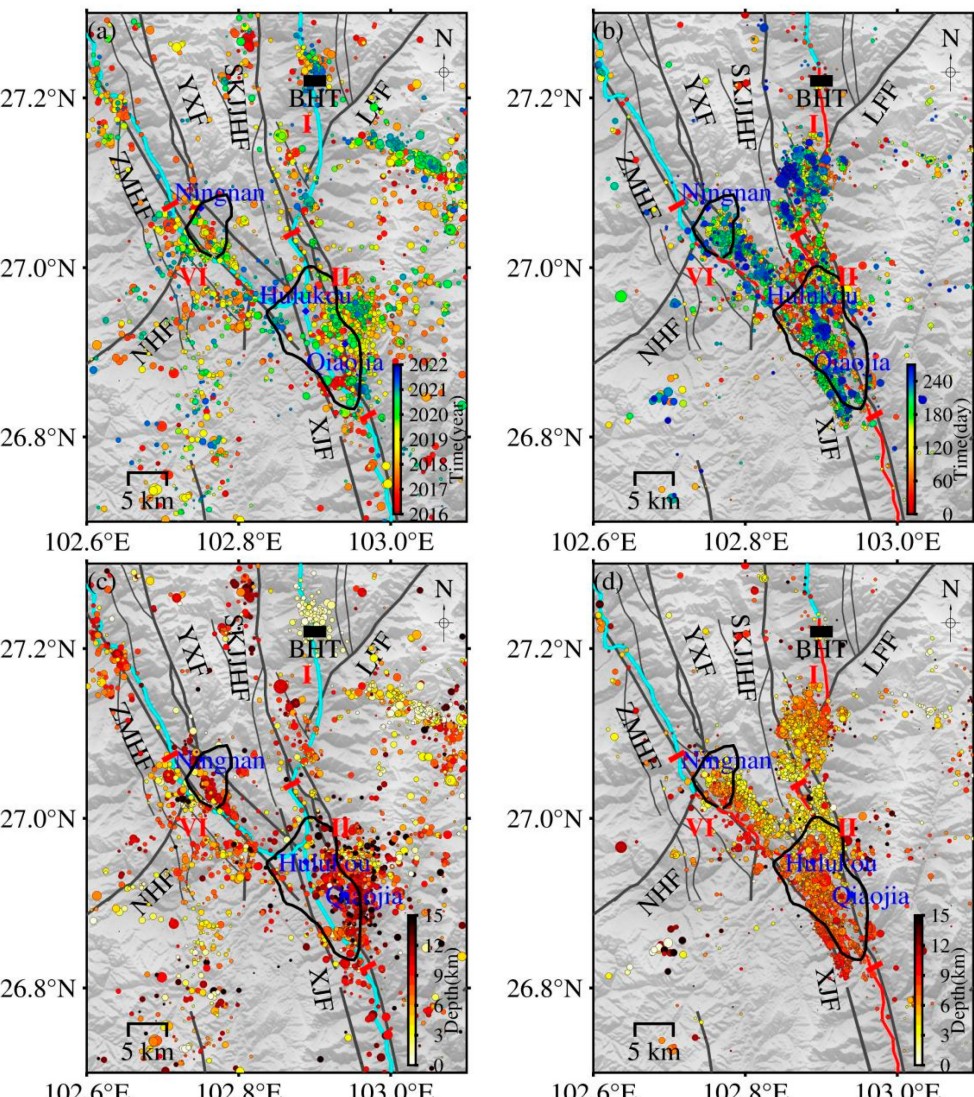

**Figure 3.** Spatial and temporal distribution of earthquakes in the reservoir area before and after impoundment. (**a**) Distribution of earthquakes that occurred before impoundment; (**b**) Distribution of earthquakes that occurred after impoundment; (**c**) Distribution of earthquakes with different depths before impoundment; (**d**) Distribution of earthquakes with different depths after impoundment. The red curves in (**b**,**d**) represent the reservoir sections where the post-impoundment water rise exceeded 30 m; black trap areas are the Qiaojia Basin and Ningnan Basin, respectively. Red short lines and Roman characters represent the I, II, and VI sections, respectively. LFF: Lianfeng fault; SKJJHF: Sikaijiaojihe fault; YXF: Yuexi fault; ZMHF: Zemuhe fault; NHF: Ninghui fault; XJF: Xiaojiang fault.

Reservoir section I (from Hulukou to the front of the dam): Before impoundment, there were few seismic activities from the Hulukou to the front of the dam, and although there were various types of focal mechanisms, their nodal plane strikes are consistent with the strike of the Sikaijiaojihe fault (Figure 5a). After one week of impoundment, the small earthquake began to increase, forming three parallel small earthquake belts extending from the Jinsha River in a northwest direction, the focal depths of the earthquakes were 3–5 km, and the largest earthquake ($M_L$3.9) occurred on 21 December 2021 in Qiluogou Town, Ningnan County, Sichuan Province. Among the 35 $M_L \geq 2$ mechanism solutions obtained (Figure 5b), the proportions of normal fault type, thrust fault type, strike-slip fault type, and other types of earthquake events account for 54.28%, 8.57%, 25.71%, and 11.44%, respectively. This suggests that in addition to the obvious dominance of normal

fault-induced earthquakes, a few small earthquakes of strike-slip and thrust types existed. Among them, the $M_L 3.9$ earthquake was the strike-slip type, and the strikes of the two nodal planes were northwest and northeast, respectively. This reservoir section is mainly composed of carbonate rocks. No large-scale karst pipeline system, karst caves, underground rivers, funnel hot springs, etc., were found in the karst hydrogeological surveys. Therefore, it is speculated that the earthquakes in this region were mainly caused by the reservoir water gravity on the shallow cracks or small-scale secondary structures or the rapid fluid penetration as the water level rose.

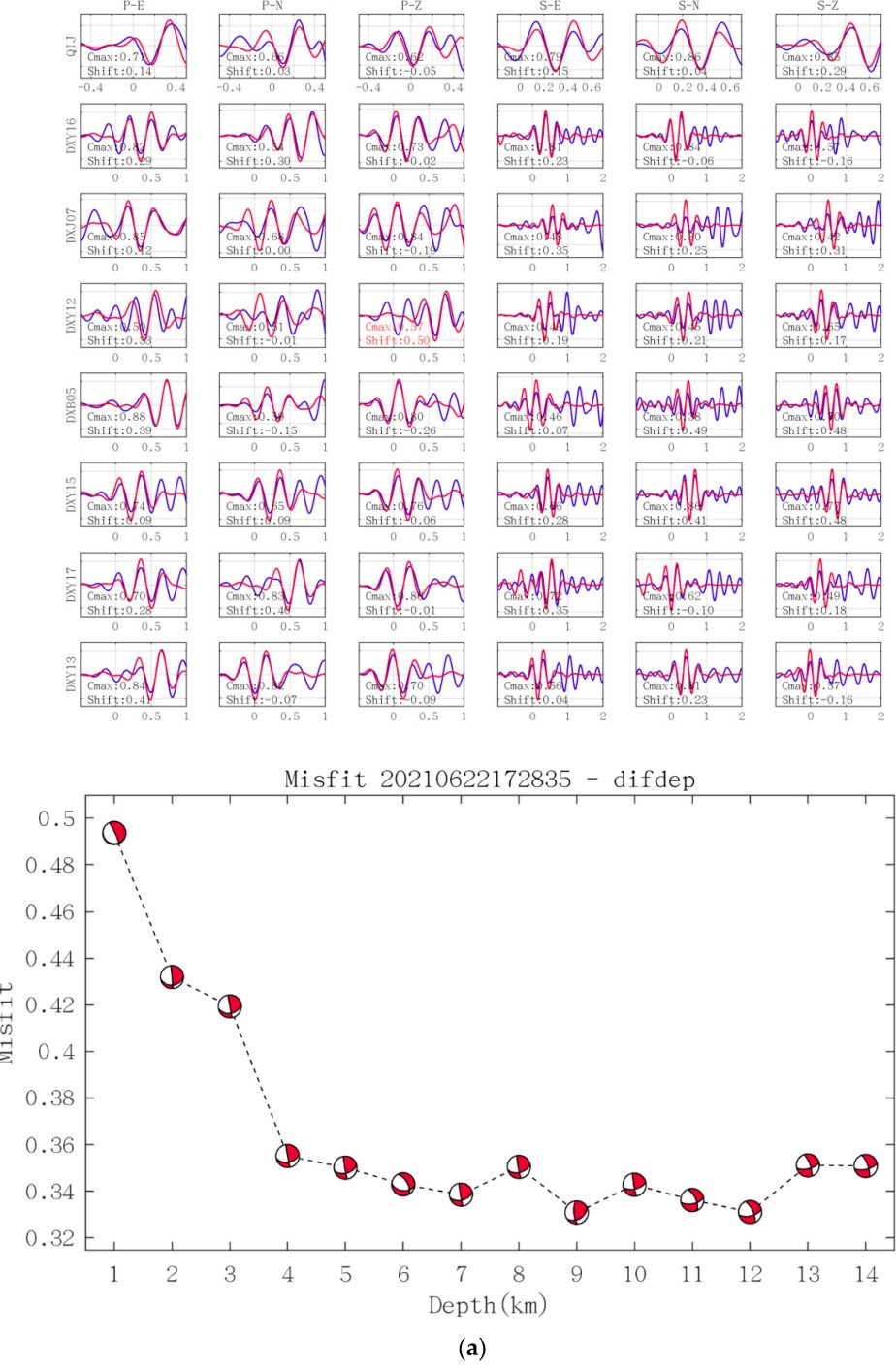

(a)

**Figure 4.** *Cont.*

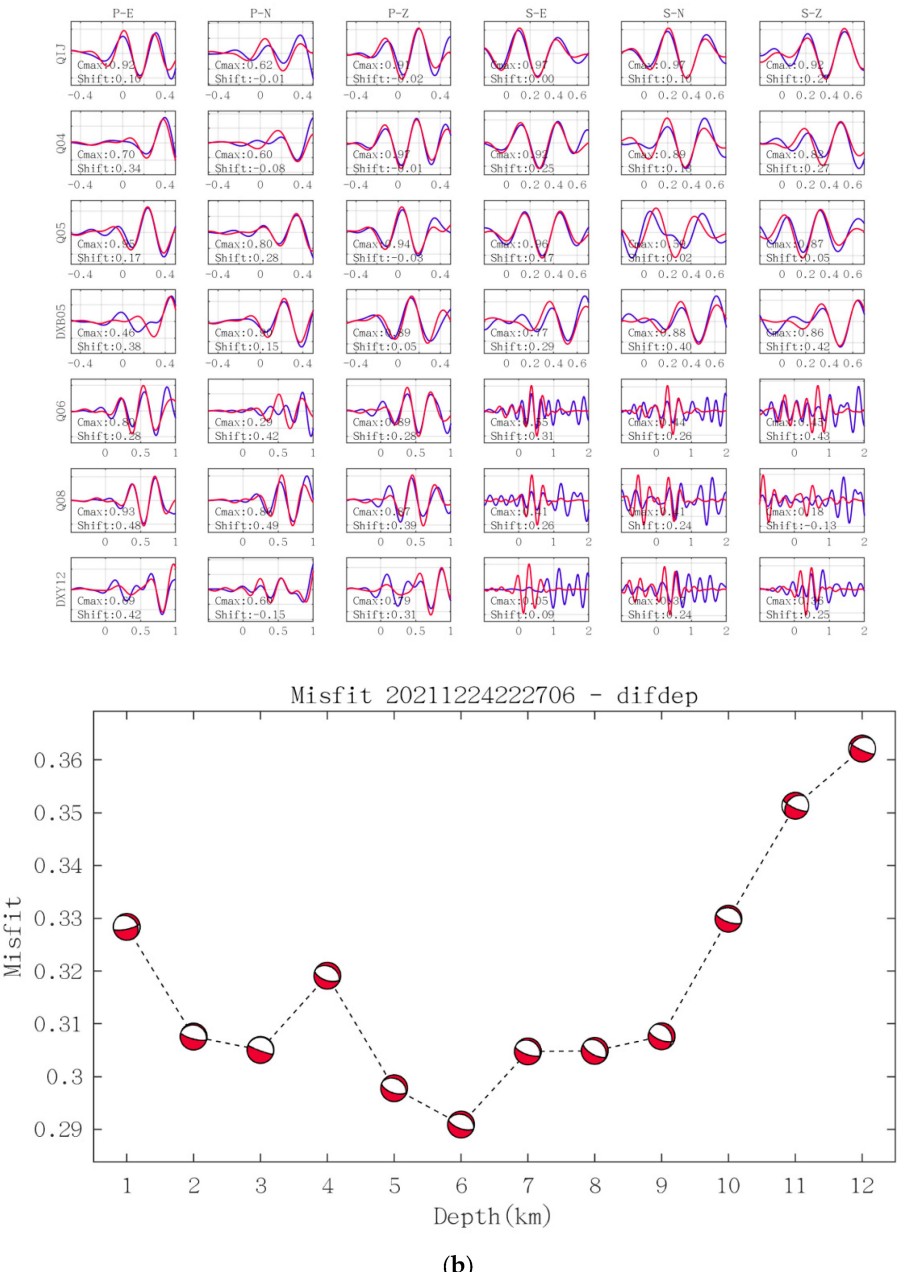

**(b)**

**Figure 4.** Focal mechanism solutions and waveform fitting. (**a**) The 22 June 2021 $M_L$ 2.2 earthquake; (**b**) The 24 December 2021 $M_L$ 3.7 earthquake. The uppermost plot in Figure 4 is the waveform matching figure. Each row is the matching condition of the observed waveform (blue) and the theoretical waveform (red) recorded by a station. From left to right, the 6 waveforms represent the EW component of the P-wave, the NS component of the P-wave, the Z component of the P-wave, the EW component of the S-wave, the NS component of the S-wave, and the Z component of the S-wave. The lower left corner shows the maximum correlation coefficient (Cmax) and the time shift (Shift). The lowermost plot shows the variation of the waveform fitting mismatch value with the change of focal depth.

Reservoir Section II (from Hulukou southward to Xiaojiang Fault): Reservoir Section II is the broadest zone in the Baihetan Reservoir area. The Xiaojiang fault is controlled by stress from the southeast and south-southeast directions of the Sichuan-Yunnan rhombic block and the northwest and northwest-west stress from the South China block [18]. The fault is a sinistral and normal fault with a north-northwest strike. Prior to the impoundment,

there were mainly seismic activities in the northern section of the Xiaojiang fault, and the earthquakes were distributed along the fault in the north-northwest direction. The mechanism solutions of 48 earthquakes (Figure 5c) were obtained, of which the normal fault type, thrust fault type, strike-slip type, and other types of seismic events account for 46.92%, 6.25%, 12.5%, and 34.33%, respectively. The largest proportion of normal fault type is related to the fact that Qiaojia is a pull-apart basin, consistent with the nature of the northern section of the Xiaojiang fault zone. After one week of impoundment, the earthquakes were distributed along the Jinsha River, Qiaojia Basin, and the northern section of the Xiaojiang fault in a south-southeast direction. The corresponding focal depths were concentrated at 0–10 km, deeper than those north of Hulukou, and showed a southward deepening pattern. Jiang et al., (2014) [19] summarized the characteristics of reservoir earthquakes in the Chinese Mainland. They found that at the initial stage of reservoir impoundment, with the development of the impoundment process, the focal depth had a gradually deepening trend, mainly related to the fluid infiltration and the gradual deepening of the depth of the changes in pore pressure. After several years, the focal depth tended to be stable. If the focal depth showed an obvious downward trend, attention should be paid to the possibility of a major earthquake. By investigating the depth change of the Koyna Reservoir before several major earthquakes, Rostogi et al., (1999) [20] found that about one month before major earthquakes, the focal depths often showed a statistically credible downward trend. After impoundment, the proportions of normal fault type, thrust fault type, strike-slip fault type, and other types of earthquakes in section II were 36.23%, 15.94%, 26.09%, and 21.74%, respectively (Figure 5d). It is prominent that although the proportion of thrust and strike-slip earthquakes increased, the proportion of normal fault earthquakes was still the largest. Particularly, this reservoir section is characterized by the most complicated geological structure and hydrogeological conditions in the entire reservoir area. The ground surface on both sides of the river valley is composed of the Quaternary fluvial sedimentary layer, and the underlying formations consist of carbonate rocks. In the Qiaojia Basin, there are several concealed water-filled faults parallel to the strike of the Xiaojiang fault. On both sides of the water-filled faults are karst and basalt water-filling bodies. Hydraulic connections may exist between the deep faults and the karst water-filling bodies. The small earthquakes distributed in belts along the Jinsha River may have been generated by the stress and strain readjustment under the physical or chemical effects (e.g., self-weight stress, reservoir water loading, wedging, and pore water pressure) of the reservoir bank slope or rock interfaces. Besides, this section is also the area with the highest seismic intensity after impoundment. In total, three $M_L \geq 3.7$ earthquakes occurred in Qiaojia, Yunnan (2021.05.24 $M_L$ 3.8, 2021.10.07 $M_L$ 3.8, 2021.12.24 $M_L$ 3.7), and the corresponding mechanism solutions were (strike 170°, dip 70°, rake −30°), (strike 345°, dip 30°, rake −70°), (strike 290°, dip 25°, rake −90°). It is suggested by the authors that the genesis of seismicity after impoundment in section II is associated with the joint effect of a regional tectonic stress field and reservoir impoundment.

　　Reservoir Section VI (from Hulukou to Nningnan along the Zemuhe fault and Heishuihe tributary): When the water level of the Heishuihe tributary rose by 30 m, seismic activity began to appear in the northwest-most of Ningnan Basin. Then, the seismic activity gradually propagated toward Hulukou in a southeast direction, forming a belt-shaped seismic distribution consistent with the north-northwest-trending fault zone basin on the east side of the Heishui River. The focal depths of the earthquakes are concentrated at 0–9 km (Figure 3d). The focal depths became shallower (about 3 km) as the earthquakes moved away from the Heishui River eastward, while those near the Heishuihe tributary and the Zemuhe fault were relatively deep (about 8 km) and the same those occurred there before loading. The seismic activity in this section of the reservoir ceased near Ningnan. Among these seismic events, the largest earthquake was the 2021.08.30 $M_L$ 3.5 earthquake in Ningnan, Sichuan. The event had a focal mechanism solution of (strike 315°, dip 65°, rake −20°) and was a strike-slip earthquake, consistent with the northwest sinistral strike-slip feature of the Zemuhe fault. Before impoundment, the proportions of normal fault

type, thrust fault type, strike-slip fault type, and other types of $M_L \geq 2$ earthquake events were 26.32%, 5.26%, 47.37%, and 21.05%, respectively (Figure 5e). The proportions after impoundment became 50.00%, 0.00%, 14.28%, and 35.72%, respectively (Figure 5f). It is obvious that the number of normal fault earthquakes in section VI increased significantly after impoundment. According to the geological background around this section, it is suggested by the authors that with the increase of the reservoir water level, water loading may lead to the increase of the maximum principal compressive stress in the vertical direction, producing small-scale normal fault activities. Meanwhile, the gravity of water and the water-induced softening of the structural surface may also promote the generation of micro-cracks in the rock mass, further causing the collapse of the free surface at the riverside. The seismic events here belong to reservoir-induced earthquakes associated with the shallow micro-cracks.

**Table 1.** Types of focal mechanism solutions.

| Type | Plunge of P-Axis $\sigma_P/°$ | Plunge of B-Axis $\sigma_B/°$ | Plunge of T-Axis $\sigma_T/°$ |
|---|---|---|---|
| Normal fault | $\geq 52$ | | $\leq 35$ |
| Normal-strike-slip | $40 \leq \sigma_P < 52$ | | $\leq 20$ |
| Strike-slip | $<40$ | $\geq 45$ | $\leq 20$ |
| Thrust-strike-slip | $\leq 20$ | | $40 \leq \sigma_T < 52$ |
| Thrust fault | $\leq 35$ | | $\geq 52$ |
| Others | | $20 < \sigma_P, \sigma_B, \sigma_T < 45$ or $40 \leq \sigma_P, \sigma_T \leq 50$ | |

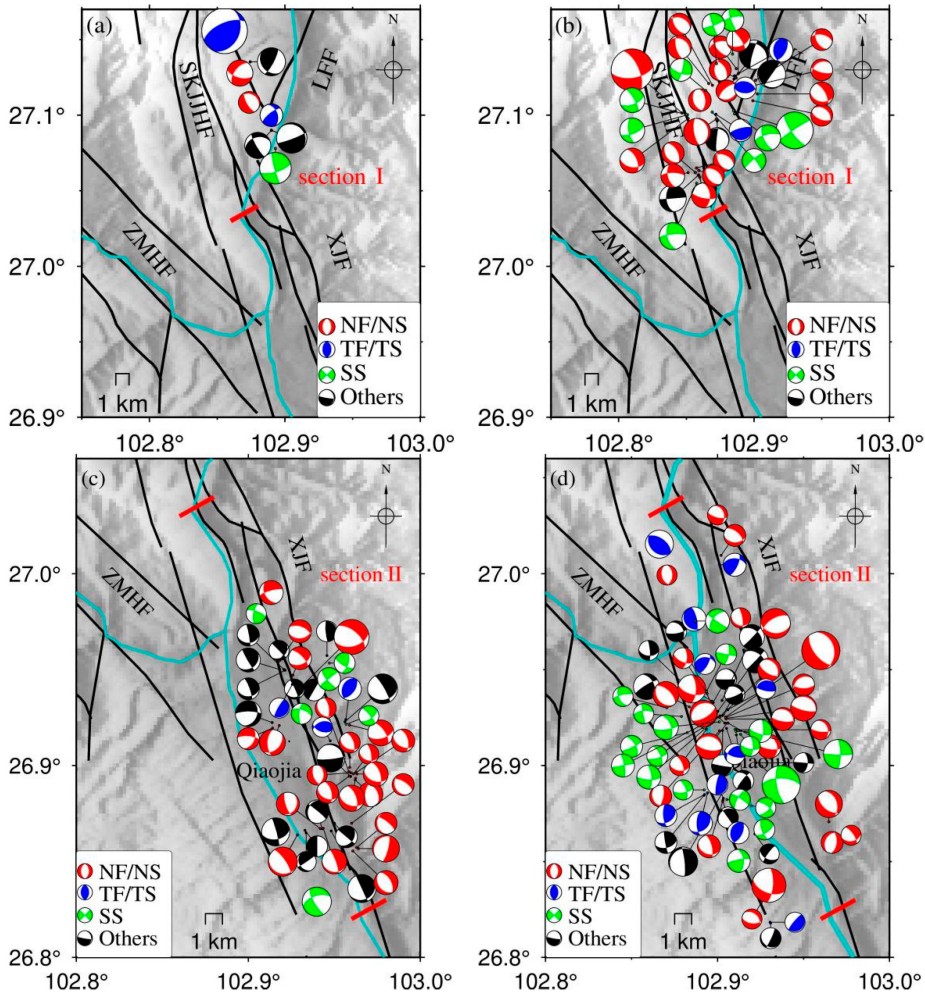

**Figure 5.** *Cont.*

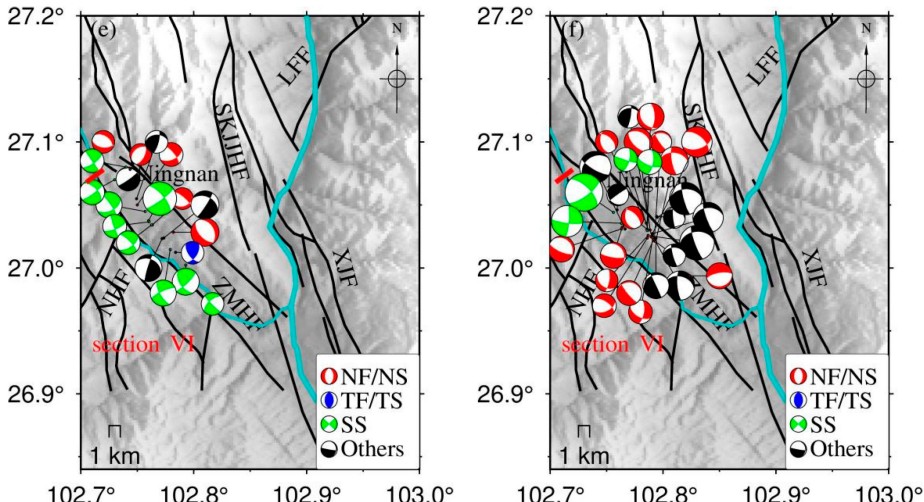

**Figure 5.** Focal mechanism solutions of $M_L \geq 2$ earthquakes around the Baihetan Reservoir. (**a,b**) Section I; (**c,d**) Section II; (**e,f**) Section VI. Left figure (**a,c,e**): Pre-impoundment (1 August 2013–5 April 2021); Right figure (**b,d,f**): Post-impoundment (6 April 2021–31 December 2021).LFF: Lianfeng fault; SKJJHF: Sikaijiaojihe fault; ZMHF: Zemuhe fault; NHF: Ninghui fault; XJF: Xiaojiang fault.

**Table 2.** Mean and standard deviation of the first 100 best solutions.

| Seismic Event | Index | Strike/° | Dip/° | Rake/° |
|---|---|---|---|---|
| 2021.06.22 $M_L$ 2.2 | Optimal solution | 175 | 70 | 55 |
| | Mean value | 173 | 79 | 42 |
| | Standard deviation | 5 | 7 | 10 |
| 2021.12.24 $M_L$ 3.7 | Optimal solution | 290 | 25 | −90 |
| | Mean value | 308 | 28 | −74 |
| | Standard deviation | 13 | 4 | 11 |

Figure 6 shows the statistics of the nodal plane strike of the earthquakes that occurred in the three reservoir sections, respectively. It is obvious that the dominant nodal plane strikes in section I are concentrated from northwest to north-northwest, consistent with the strike of the nearest large-scale fault zone (Sikaijiaojihe fault). The dominant nodal planes of the mechanism solution of earthquakes in section II are concentrated from northwest to north-northeast, which accords well with the strike of the Xiaojiang fault. For section VI, the dominant nodal plane strikes are also concentrated in the direction from northwest to north-northeast, consistent with the strike of the Zemuhe fault. The overall pattern of the focal mechanism solutions indicates that most of the microearthquakes occurred near the river channel after impoundment and were generally distributed along valley basins controlled by large fault zones or smaller-scale structures.

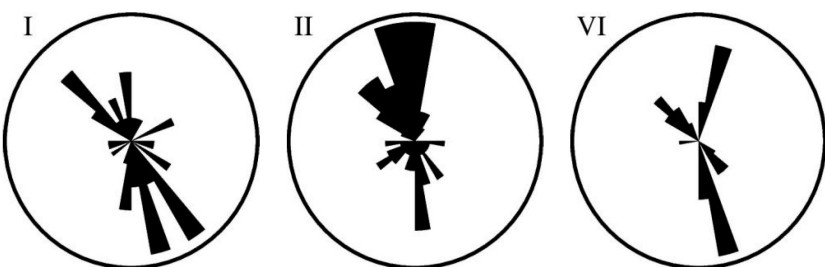

**Figure 6.** Rose diagrams of nodal strikes for focal mechanism in three regions of Baihetan Reservoir.

### 4.3. Stress Field

Furthermore, the iterative joint inversion method [21] was used to invert the stress field of the three sections. The pre- and post-impoundment P and T -axes of the earthquake and the principal stress axis added with 100 times of random noise are shown in Figure 7 and Table 3, respectively. Based on the focal mechanism solutions of small earthquakes in the northern section of the Xiaojiang fault, Yan (2015) [22] found that the azimuth of the maximum principal compressive stress $\sigma_1$ axis in this area was around 120°, and the dip was 23°. The results also showed that the azimuth of the maximum tensile stress $\sigma_3$ axis was 215°, and the dip was around 14°. The azimuth of the medium principal compressive stress $\sigma_2$ axis was around 335°, and the dip was around 63°. The above studies indicate that this area was mainly affected by near-horizontal compressive stress and tensile stress before the impoundment of the Baihetan Reservoir. According to Figure 7 and Table 3, the post-impoundment azimuth and dip angle of the maximum tensile stress axes of the three reservoir sections are almost in the same south-southwesterly direction, and nearly horizontal. The maximum principal compressive stress axes are all nearly vertical, revealing that the strong vertical principal compressive stress and the horizontal tensile stress field control both sides of the river. The R-value is expressed as R = $(\sigma_2-\sigma_1)/(\sigma_3-\sigma_1)$, indicating that the post-impoundment stress field of the three sections tends to be dominated by tensile stress. Among the three sections, section I had the highest water level rise in front of the dam. The earthquakes in section VI before impoundment were mainly distributed along the Zemuhe fault. The earthquakes after impoundment generally occurred in the Zemuhe fault zone and the Ningnan Basin on the east side of the Heishui River.

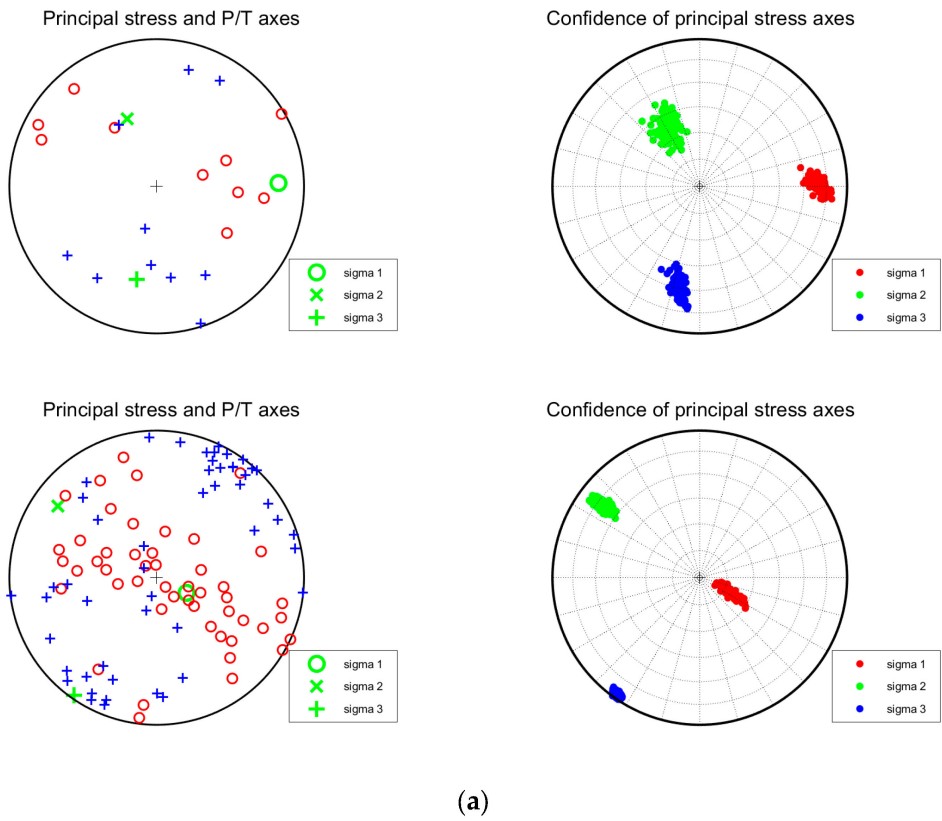

(**a**)

**Figure 7.** *Cont.*

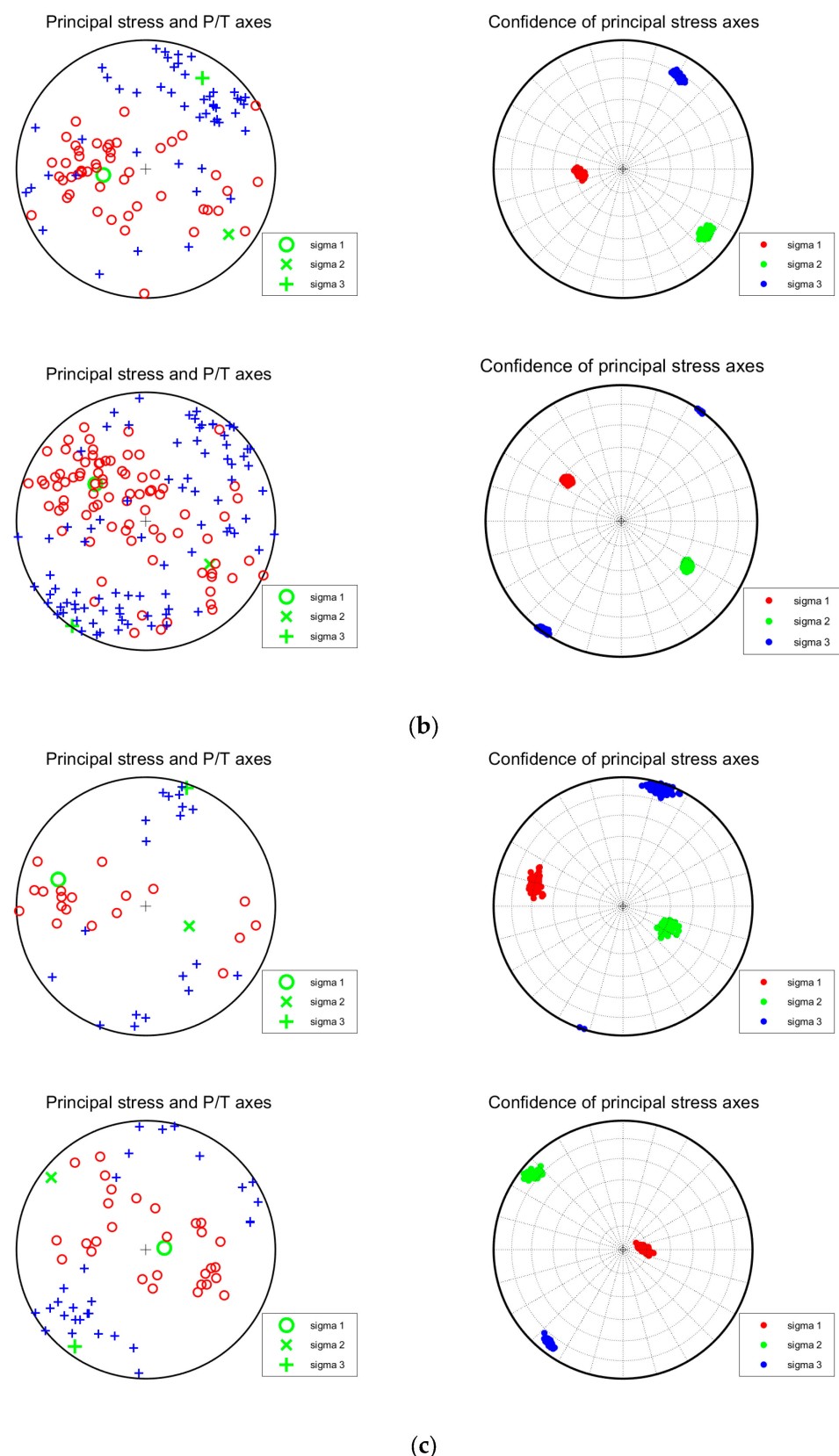

**Figure 7.** Spatial distribution map of P, T, and stress axes in three sections of the Baihetan Reservoir area after impoundment. (**a**) Section I; (**b**) Section II; (**c**) Section VI.

**Table 3.** Number of focal mechanisms and the inversion results of the stress field in each reservoir section before and after impoundment.

| Section | Stress Field | $\sigma_1(/°)$ Azimuth | Dip | $\sigma_2(/°)$ Azimuth | Dip | $\sigma_3(/°)$ Azimuth | Dip | R | N |
|---|---|---|---|---|---|---|---|---|---|
| I | Before impoundment | 88.5 | 18.4 | 336.3 | 48.6 | 192.2 | 35.5 | 0.58 | 8 |
| | After impoundment | 116.9 | 71.5 | 305.9 | 18.3 | 215.0 | 2.7 | 0.26 | 35 |
| II | Before impoundment | 262.0 | 62.8 | 128.5 | 19.5 | 31.8 | 18.2 | 0.60 | 48 |
| | After impoundment | 306.4 | 49.8 | 124.2 | 40.2 | 215.1 | 1.1 | 0.55 | 69 |
| VI | Before impoundment | 287.0 | 29.9 | 114.6 | 59.9 | 18.9 | 3.3 | 0.66 | 19 |
| | After impoundment | 84.5 | 78.4 | 307.5 | 8.5 | 216.3 | 7.8 | 0.55 | 28 |

Overall, the Baihetan Reservoir is currently in the initial stage of impoundment. The considerable rise in seismicity frequency is related to the high water level and its drastic change following 6 April 2021. Additionally, a normal fault is the most dominant focal mechanism solution of the earthquakes, similar to what happened in the Three Gorges Reservoir and Xiluodu Dam within 1 year of impoundment.

## 5. Conclusions

As of 31 December 2021, the water level in Hulukou town had risen by nearly 158 m. In this study, the high-frequency waveform matching method is employed, and the focal mechanism solutions of 207 $M_L \geq 2.0$ earthquakes are obtained in the vicinity of the Baihetan Dam. By combining seismicity, focal mechanism solutions, regional geological settings, and the impoundment-induced water-level change, the major conclusions are obtained as follows:

(1) The distribution of pre-impoundment earthquakes in the reservoir area is mainly controlled by regional stress fields and faults. As the impoundment proceeds, the number of earthquakes increases significantly, especially small and microearthquakes. These earthquakes are clustered in Hulukou town and dominantly distributed along the north-northeastern Sikaijiaoji and Jinsha Rivers, the north-northwestern Zemuhe fault and Heishuihe tributary, and the south-southeastern Xiaojiang fault and Jinsha River, showing an overall "Y-shaped" pattern. In addition, the number of earthquakes in the Ludian aftershock zone enormously decreases;

(2) After the impoundment, microearthquakes are observed in three reservoir sections, where normal fault-induced earthquakes are the most predominant. The dominant distribution of the fracture planes obtained from the inversion of the mechanism solution is consistent with the direction of the local main structures or fault zones, indicating that they are controlled by the local tectonic environment;

(3) In the three reservoir sections with post-impoundment microearthquakes, the azimuth angle and dip angle of the maximum tensile stress axis are consistent; both are in the south-southwestern direction and nearly horizontal. whereas the maximum primary compressive stress axes are nearly vertical, suggesting the effect of post-impoundment vertical compressive stress and horizontal tensile stress fields in the areas with microearthquakes;

(4) Pre-existing fissures and structures in the reservoir area are the prerequisites for inducing earthquakes, and the water level change is an essential external factor that influences earthquake occurrence;

(5) The occurrence of earthquakes is closely related to the drastic increase in impoundment loading and the water-level-change-induced elastic stress on the side slope, joints, fissures, and other small-scale structures. Such features accord well with the features observed in the pre-impoundment period of the downstream area of the Xiluodu Reservoir. Due to the continuous influence of reservoir impoundment on

the surrounding geological environment, and considering complex regional tectonic structures and the occurrence of strong historical earthquakes, it is necessary to pay close attention to the possibility of moderate-to-strong earthquakes in the reservoir area in the future.

**Author Contributions:** Conceptualization, C.Z.; methodology, W.G.; software, W.G.; validation, W.G. and C.Z.; formal analysis, C.Z.; investigation, W.G.; resources, C.Z.; data curation, W.G.; writing—original draft preparation, WG.; writing—review and editing, C.Z.; visualization, W.G. and C.Z.; supervision, C.Z.; project administration, C.Z.; funding acquisition, C.Z. All authors have read and agreed to the published version of the manuscript.

**Funding:** This research was funded by the National Key Research and Development Program of China (2021YFC3000703) and the Special Fund of the Institute of Earthquake Forecasting, China Earthquake Administration (CEAIEF20220401).

**Data Availability Statement:** The data presented in this study are not publicly available due to confidentiality agreements.

**Acknowledgments:** The authors would like to thank Li Junlun from the University of Science and Technology of China provided a program for the inversion of earthquake focal mechanism solutions. The topography data used in this study was obtained from SRTMGL3 (https://lpdaac.usgs.gov/products/srtmgl3v003/ (accessed on 22 June 2021)), and figures were plotted using GMT [23]. Also, we would like to express our appreciation to the reviewers of this paper for their valuable comments.

**Conflicts of Interest:** The authors declare no conflict of interest.

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
