# Peer review of "Focal Mechanisms and Stress Field Characteristics of Microearthquakes in Baihetan Reservoir in the Downstream Area of Jinsha River"

_water, doi:10.3390/w15040709_

Round 1

Reviewer 1 Report

 the manuscript under the title “Focal Mechanisms and Stress Field Characteristics of Microearthquakes in Baihetan Reservoir in the Downstream Area of Jinsha River” by the authors is a classical article in earthquake focal mechanism solutions.

I suggest publishing this manuscript after major revisions: 

 1-      Please move this sentence from the abstract   “The Baihetan hydropower station, which is situated in the downstream area of Jinsha river, has a height of 289 m and a reservoir storage capacity reaching as high as 2.0627 x 1010 m3. It is the second largest hydropower station in the world (ranked only second to the Three Gorges station)” to the introduction section  or deletes it as it is not important in the abstract furthermore, the abstract is very long; other sentences should also be moved to the results section.

 2-      Please try to rewrite the abstract again. The abstract in its current form is similar to the results and conclusion sections. The ideal abstract should include some sentences in the case study, methods and methodology, results and conclusion,

 3-      In figure 1 you have two panels, the upper panel (a) and the lower panel (b), you should in the figure caption specify the figure panels what are features in each panel and detect and locate the location of the hydropower station.

 4-      What about stress drop values and other seismic source parameters? Did you determine the values of stress drop? I suggest that you read this article “Characteristics of earthquakes recorded around the High Dam Lake with comparison to natural earthquakes using waveform inversion and source spectra. Pure Appl. Geophys. doi.org/10.1007/s00024-020-02490-4”.

 5-      In figure 3 the size of the symbols of earthquakes are very small. Try to increase the size of the symbols

 6-      Please separate “Discussion and Conclusions” into two sections. 

Author Response

Response to Reviewer 1 Comments

Point 1: Please move this sentence from the abstract “The Baihetan hydropower station, which is situated in the downstream area of Jinsha river, has a height of 289 m and a reservoir storage capacity reaching as high as 2.0627 x 1010 m3. It is the second largest hydropower station in the world (ranked only second to the Three Gorges station)” to the introduction section or deletes it as it is not important in the abstract furthermore, the abstract is very long; other sentences should also be moved to the results section.

Response 1: the sentence has been deleted and the abstract simplified.

Point 2: Please try to rewrite the abstract again. The abstract in its current form is similar to the results and conclusion sections. The ideal abstract should include some sentences in the case study, methods and methodology, results and conclusion,

Response 2: the abstract has been rewrited.

Point 3: In figure 1 you have two panels, the upper panel (a) and the lower panel (b), you should in the figure caption specify the figure panels what are features in each panel and detect and locate the location of the hydropower station

Response 3: the figure caption has been modified, and the location of the hydropower station has been detected and located.

Point 4: What about stress drop values and other seismic source parameters? Did you determine the values of stress drop? I suggest that you read this article “Characteristics of earthquakes recorded around the High Dam Lake with comparison to natural earthquakes using waveform inversion and source spectra. Pure Appl. Geophys. doi.org/10.1007/s00024-020-02490-4”.

Response 4: The idea of this article is worthy of reference, but this paper mainly deals with a large number of small earthquake mechanism solutions, and studies the influence of water storage on stress field. The study of source spectrum parameters is still in progress.

Point 5: In figure 3 the size of the symbols of earthquakes are very small. Try to increase the size of the symbols.

Response 5: the size of the symbols in figure 3 has been increased.

Point 6: Please separate “Discussion and Conclusions” into two sections.

Response 6: The content of the previous “Discussion and Conclusions” section is the conclusion, so “Discussion and Conclusions” has been changed to “Conclusions” and the content of the conclusions has been revised.

Reviewer 2 Report

Detailed comments for the authors can be found inside the attached PDF file.

Round 2

Reviewer 1 Report

the manuscript now ready to publish